# Effects of Developmental Arsenic Exposure on the Social Behavior and Related Gene Expression in C3H Adult Male Mice

**DOI:** 10.3390/ijerph16020174

**Published:** 2019-01-09

**Authors:** Soe-Minn Htway, Mya-Thanda Sein, Keiko Nohara, Tin-Tin Win-Shwe

**Affiliations:** 1Department of Physiology, University of Medicine, Magway, Magway 04011, Myanmar; drsmhtway@gmail.com (S.-M.H.); dr.myathandasein@gmail.com (M.-T.S.); 2Center for Health and Environmental Risk Research, National Institute for Environmental Studies, Tsukuba 305-8506, Japan; keikon@nies.go.jp

**Keywords:** arsenic, developmental, gene expression, mice, prefrontal cortex, social behavior

## Abstract

Arsenic is carcinogenic and teratogenic. In addition, it is also a developmental neurotoxicant. Little is known however about the effect of arsenic exposure during brain development on social behavior. This study aimed to detect the effect of developmental arsenic exposure on social behavior and related gene expression in C3H adult male mice. Pregnant C3H mice were exposed to sodium arsenite (NaAsO_2_, 85 ppm in the drinking water) from gestational day (GD) 8 to 18. The F1 generation male pups from different mothers were taken and social behavior tasks were examined. Social behavioral-related gene expression in the prefrontal cortex was determined by the real-time RT-PCR method. The mice with developmental arsenic exposure showed poor sociability and poor social novelty preference. Glutamate receptor expression (NMDA and AMPA receptor subunits) showed no significant difference, but gene expressions of serotonin receptor 5B (5-HT 5B) and brain-derived neurotrophic factor (BDNF) were significantly decreased (*p* < 0.05) in the arsenic-exposed group compared to control group. The heme oxygenase-1 (HO-1) and cyclooxygenase-2 (COX-2) gene expressions were not significantly different. Our findings indicate that developmental arsenic exposure might affect social behavior by modulating serotonin receptors and reducing BDNF. Some oxidative stress markers and inflammatory markers were not affected.

## 1. Introduction

Arsenic is a naturally occurring metalloid and can be found in water as the inorganic form (arsenite or arsenate) [1]. Arsenic contamination in drinking water is a worldwide problem, especially in Bangladesh. According to the study of Ahmad et al., 50 million people in Bangladesh were exposed to arsenic-contaminated water (above 0.05 mg/L) and the highest level of contamination is 2.97 mg/L [2]. In 2012, it was estimated that about 19 million people in Bangladesh were still exposed to arsenic concentrations above the national standard of 0.05 mg/L. According to the World Health Organization provisional guideline value (0.01 mg/L), about 39 million people in Bangladesh were exposed to arsenic-contaminated water [3]. There are several health problems related to arsenic exposure; skin lesions, cardiovascular disease, neurological disorders, and diabetes [4,5,6]. Arsenic is a well-known carcinogen and it has been widely studied as a human carcinogenic agent. Arsenic has also embryotoxicity and teratogenicity effects [7,8].

Arsenic can cross the placenta and it can cause neural tube defects. It can also cause delay in growth and development of the fetus [9]. Both inorganic and methylated arsenicals accumulate in many parts of the brain, especially in the pituitary [10]. There are some studies related to the neurobehavioral effects of arsenic. They pointed out that chronic ingestion of arsenic can be related to decreased intelligence, behavioral deficits including learning, memory and locomotion, and mood disorders like depression [11,12,13]. Epidemiological studies have also indicated that exposure to arsenic in drinking water in utero or early childhood resulted in the increase of mortality in young adults by bladder cancer, laryngeal cancer, lung cancer and chronic renal disease [14,15,16]. There are different mechanisms of arsenic-induced neurotoxicity which include altered glucocorticoid signaling, cholinergic, glutamatergic and monoaminergic signaling, neurogenesis and other forms of synaptic plasticity [17,18].

Rodriguez et al. indicated that rats exposed to arsenic during development present deficits in spontaneous locomotor activity and alterations in a spatial learning task [19]. Martinez et al. suggested that perinatal arsenic exposure may disrupt the regulatory interactions between the hypothalamic-pituitary-adrenal axis and the serotonergic system in the dorsal hippocampal formation predisposing to depressive-like behavior in mice [20]. Tyler and Alan reported that perinatal arsenic exposure altered the hippocampal morphology and neurogenesis-related gene expression in mice. They believed that the lack of differentiated neural progenitor cells necessary for hippocampal-dependent learning leads to behavioral deficits observed during adulthood [21]. Aung et al. demonstrated that prenatal arsenic exposure leads to behavioral inflexibility to changing tasks in adulthood and cortical disarrangement in the prelimbic cortex in C3H mice [22].

Cumulative evidence indicates that arsenic is a developmental neurotoxicant, affecting intellectual functions, even at low levels. The developing brain seems to be the most vulnerable to neurotoxic effects of arsenic. Prenatal and early postnatal arsenic exposure linked to the reduction in brain weight, number of glia and neurons, and alterations in neurotransmitter systems. Oxidative stress may be a mechanism of arsenic neurotoxicity [13]. The mechanisms of developmental neurotoxicity and behavioral alterations induced by arsenic exposure still need to be explored. Little is known about the effect of arsenic exposure during brain development on social behavior. Social behavior is related to learning and memory [23]. It is also influenced by mood changes [24,25]. Most of the previous studies investigated the changes in the hippocampus [20,21,26,27,28]. The prefrontal cortex is responsible for moderating social behavior [29]. This study aimed to detect the effect of developmental arsenic exposure on social behavior and related gene expression in the prefrontal cortex of C3H adult male mice. In this study, our genes of interest are that of glutamate receptors, brain-derived neurotrophic factor (BDNF), and serotonin receptor which are related to synaptic plasticity, learning, synaptogenesis and mood changes [26,30,31].

## 2. Materials and Methods

### 2.1. Experiment 1

#### 2.1.1. Animals

Pregnant C3H mice were purchased from JAPAN SLC (Shizuoka, Japan) and kept at 24 ± 1 °C and 12 h light/dark cycle with free access to water and food. The subject mice were exposed to sodium arsenite (NaAsO_2_, 85 ppm (85 mg/L) in the drinking water) from gestational day (GD) 8 to 18. The control mice were given tap water. The weight of water bottle for each mouse was measured before and after providing *ad libitum* access to water. The reasons for selection of a 85 ppm dose of NaAsO_2_ are: (1) species differences were observed between human and mouse and a high dose is required for detection of neurotoxic effects in C3H mice and no maternal toxicity and teratogenicity were observed at the dose of present study; (2) this is a standard dose to detect the effects of developmental exposure to arsenic on neurotoxicity, reproductive toxicity and carcinogenicity in our research group [22,32,33]. The F1 male pups were weaned at postnatal day 21 and were randomly selected from different dams and housed under the same condition as the dams (*n* = 3 per cage). At the age of 70-week, male pups were housed in the individual cage. The reasons for using only male pups are (1) there is a sex difference in susceptibility to arsenic; (2) there is the estrous cycle in female and possible hormonal effects on social behavior [19]. The experimental protocols were approved by the Ethics Committee of the Animal Care and Experimentation Council of the National Institute for Environmental Studies (NIES), Japan (Ethics approval code number: 13-040).

#### 2.1.2. Social Behavior

Social behavior task was examined at the age of 74-week using a three-chambered social behavior apparatus (Muromachi Kikai Co. Ltd., Tokyo, Japan) and the ANY-maze software (Stoelting Co., Wood Dale, IL, USA). Three-chambered Plexiglas box (100 cm × 100 cm × 35 cm) purchased from Muromachi Kikai was used as described previously [34]. The box was divided into three equal chambers by transparent partition with a square-shaped hole (10 cm × 10 cm) at the base, allowing free access to all chambers. Social behavior test consists of three phases including a habituation phase for 5 min, a sociability phase for 10 min and a social novelty preference for 10 min. First of all, a subject mouse was placed in the middle chamber and allowed 5 min for habituation while an empty wired cup (15 cm in diameter and 30 cm height) was placed in each side chamber. Then, a stranger rat (Stranger 1, age-matched) was placed in the wired cup of any side chamber and the other was left empty. The subject mouse was allowed to explore for 10 min. Time spent by the subject mouse in each chamber was video-recorded and its sociability was analyzed by computer software. For social novelty preference test, another stranger (Stranger 2, age-matched) was placed in the wired cup of the other side chamber. Time spent by the subject mouse for each stranger (Stranger 1 and 2) was recorded for 10 min and then analyzed. The time spent with its head facing the cup from a distance of within 1 cm was regarded as time spent for this cup.

#### 2.1.3. Light-Dark Transition Test

For light-dark transition, a subject mouse was placed in a clear plastic box (40 cm × 20 cm × 25 cm), one half of which was a dark chamber (20 cm × 20 cm × 25 cm) and the other half was a light chamber (20 cm × 20 cm × 25 cm) which was illuminated by a 40-W bulb (about 350 lux on the floor). After 1 h habituation in the test room, mice were gently placed to the corner of the dark side away from the doorway. There was an open doorway (2 cm × 5 cm) between two chambers. Time spent for each chamber (light or dark), total moving time, number of transitions between light and darkness and the latency to the first entry to the light chamber were measured and analyzed by ANY-maze software (Stoelting Co., Wood Dale, IL, USA).

#### 2.1.4. Quantification of Social Behavior-Related Gene Expression

In the present study, for Experiment (1), after completion of behavior test, the 74-week-old male mice were sacrificed under deep pentobarbital anesthesia and the prefrontal cortex collected for the gene expression assay. The prefrontal cortex of each mouse was dissected and frozen in liquid nitrogen and then stored at −80 °C. Total RNA extraction was performed by EZ-1 RNA tissue mini kits and BioRobot EZ-1 (Qiagen GmbH, Hilden, Germany) as described previously [34,35,36,37,38,39]. Purity and concentration of total RNA was examined by ND-1000 NanoDrop RNA Assay protocol (NanoDrop Technologies, Wilmington, DE, USA). Then, first strand cDNA synthesis from total RNA was done by SuperScript RNase H-Reverse Transcriptase II (Invitrogen, Carlsbad, CA, USA) and thermal cycler (Gene Atlas E, ASTEC, Fukuoka, Japan ). The mRNA expression levels of 18S rRNA (internal control), N-methyl-D-aspartate (NMDA) receptor subunits (NR1, NR2A and NR2B) and α-3-hydroxy-5-methyl-4-isoxazole propionic acid (AMPA) receptor subunits (GluR1, GluR2, GluR3 and GluR4) were determined by real-time RT-PCR method using Applied Biosystems (ABI) Prism 7000 Sequence Detection System (Applied Biosystems Inc., Foster City, CA, USA). The primers for NR1, NM_008169; NR2A, NM_008170; NR2B, NM_008171; GluR1, NM_008165; GluR2, NM_013540; GluR3, NM_016886 and GluR4, NM_019691 were purchased from Qiagen Sample & Assay Technologies (Qiagen GmbH, Hilden, Germany). The primers for 18S rRNA (forward: 5′-TACCACATCCAAAAGGCAG-3′, reverse: 5′-TGCCCTCCAATGGATCCTC-3′) was purchased from Hokkaido System Science (Hokkaido, Japan). Data were analyzed by using comparative threshold cycle method. The relative mRNA expression levels were expressed as mRNA signals per unit of 18S rRNA expression.

### 2.2. Experiment 2

The previous study showed that behavioral inflexibility and cortical disarrangement in the prelimbic cortex was observed in 60-week-old C3H male mice after prenatal exposure to 85 ppm arsenic [22]. This finding prompted us to examine social behavior after the same exposure period (GD 8 to 18). Therefore, we performed the same exposure design and examined social behavior and glutamate receptor NMDA and AMPA in the prefrontal cortex in 74-week-old male C3H mice (Experiment 1 in the present study). We found social behavior impairment but no NMDA and AMPA receptor alteration in the prefrontal cortex in 74-week-old male C3H mice. We suggested that social behavior impairment may be due to social behavior-related genes such as serotonin, inflammatory cytokines and oxidative stress in the brain other than glutamate receptors. Therefore, we aimed to study the effect of gestational exposure to 85 ppm arsenic on social behavior and their related gene expression in the prefrontal cortex in 15–17-week-old C3H male mice. Unfortunately, we could not perform social behavior test and only examined social behavior related gene expression in the prefrontal cortex. Pregnant C3H mice were purchased from CLEA Japan (Tokyo, Japan). The experimental protocols were approved by the Ethics Committee of the Animal Care and Experimentation Council of the National Institute for Environmental Studies (NIES), Japan (Ethics approval code number: AE-15-02).

#### 2.2.1. Quantification of Social Behavior-Related Gene Expression

The F1 mice (*n* = 7) were sacrificed under deep pentobarbital anesthesia at the age of 15–17 week. The prefrontal cortex of each mouse was dissected and frozen in liquid nitrogen and then stored at -80 °C. Total RNA extraction was performed by EZ-1 RNA tissue mini kits and EZ-1 Advance XL (Qiagen). Purity and concentration of total RNA was examined and cDNA synthesis from total RNA was done as the experiment (1). The mRNA expression levels of 18S rRNA (internal control), NMDA receptor subunits (NR1, NR2A and NR2B) and AMPA receptor subunits (GluR1, GluR2, GluR3 and GluR4), 5-hydroxytryptamine (serotonin) receptor 5B (5-HT 5B) and brain-derived neurotrophic factor (BDNF) were determined by using real-time RT-PCR (Light Cycler 96, Roche, Germany). The primers for NR1, NM_008169; NR2A, NM_008170; NR2B, NM_008171; GluR1, NM_008165; GluR2, NM_013540; GluR3, NM_016886; GluR4, NM_019691; 5- HT 5B, NM_024395 and BDNF, NM_012513 were purchased from Qiagen Sample & Assay Technologies. The primers for 18S rRNA (forward: 5′-TACCACATCCAAAAGGCAG-3′, reverse: 5′-TGCCCTCCAATGGATCCTC-3′) was purchased from Hokkaido System Science. Data were analyzed as above.

#### 2.2.2. Quantification of Oxidative Stress and Inflammatory Markers

The mRNA expression levels of heme oxygenase-1 (HO-1) and cyclooxygenase-2 (COX-2) were determined by using real-time RT-PCR (Light Cycler 96, Roche). The primers for HO-1, NM_010442, and COX-2, NM_011198 were purchased from Qiagen Sample and Assay Technologies. Data were analyzed as above.

### 2.3. Statistical Analysis

All the data were presented as mean±standard error (SE). The statistical analyses were performed using the StatMate II statistical analysis system for Microsoft Excel, Version 5.0 (Nankodo Inc., Tokyo, Japan). Paired *t* test was used to analyze the exploration time to empty cup and Stranger 1, followed by Stranger 1 and Stranger 2. Time spent in light side and messenger RNA levels were analyzed by Student’s *t* test. Difference at *p* < 0.05 was regarded as significant.

## 3. Results

### 3.1. Effect of Developmental Arsenic Exposure on Social Behavior

In control mice, time spent for Stranger 1 was significantly greater than that for empty cup. In contrast, mice with developmental arsenic exposure did not spend time differently for Stranger 1 and empty cup, indicating that they had decreased sociability (Figure 1A, * *p* < 0.05). For social novelty preference test, the arsenic-exposed mice showed the preference to Stranger 2, but it was not statistically significant while the control mice spent significantly more time for Stranger 2 (Figure 1B, * *p* < 0.01). Time spent in the light side was not significantly different between two groups (*n* = 14). However, mice in both groups spent more time in the light side than in the dark side (Figure 1C).

### 3.2. Effect of Developmental Arsenic Exposure on Prefrontal Cortex Expression of Social Behavior-Related Genes

Regarding gene expression of NMDA receptor subunits (NR1, NR2A and NR2B) and AMPA receptor subunits (GluR1, GluR2, GluR3 and GluR4), there was no significant difference between the control and arsenic-exposed groups in both experiment 1 (*n* = 14) (Figure 2A,B) and the experiment 2 (*n* = 7) (Figure 3A,B). The 5-HT 5B and BDNF expressions were significantly decreased in the arsenic-exposed group compared to the control group (*n* = 7) (Figure 4A,B, * *p* < 0.05).

### 3.3. Effect of Developmental Arsenic Exposure on Prefrontal Cortex Expression of Oxidative Stress and Inflammatory Marker Genes

The oxidative stress marker HO-1 gene expression was not significantly different between control and arsenic-exposed groups (*n* = 7) (Figure 4C). The potent inflammatory marker COX-2 mRNA tended to decrease in arsenic-exposed group, but not statistically significant (*n* = 7) (Figure 4D).

## 4. Discussion

The major findings of this study were that the adult male mice showed poor sociability to the Stranger 1 (sociability test) and also to the Stranger 2 (social novelty preference test). It indicated that developmental arsenic exposure impaired the sociability and social novelty preference in F1 mice. It is in accordance with the previous study done by Rodriguez et al. in which both prenatal (GD15) and postnatal (PND1) exposure of arsenic (36.7 ppm) for 4 months resulted in alterations in maturation process and behavior. Their study indicated that even low dose exposure (starting from PND1 to the age of 17 weeks) resulted in increased number of errors in a delayed alternation task in comparison to the control group [19]. Aung et al. also reported that gestational exposure (GD 8 to 18) to 85 ppm NaAsO_2_ resulted in increased discrimination error rate in spatial learning activity tested by IntelliCage accompanied with neuronal morphology changes in 60-week-old male mice [22]. According to these studies, it could be recognized that behavioral alterations can be seen in both young adult (17-week-old) mice and old adult (60-week-old) mice whatever the low (36.7 ppm) or high (85 ppm) dose of arsenic exposure was given in prenatal (GD8-GD18) or postnatal (PND1-PND120) or perinatal (GD15-PND120) period (Table 1). In our study, behavior task was done only in the experiment 1 (74-week-old mice) and impaired social behavior was found. We supposed that impaired social behavior would be also found in the experiment 2 (15–17-week-old mice) if we had a chance to do the behavior task.

In this study, C3H mice were used because the C3H mice are good for behavior assessment and neurochemical analysis [40]. It was reported that early life arsenic exposure in rodents can affect higher brain functions permanently through later life. Critical periods may be gestational, perinatal and early childhood [22,41]. Investigation of F1 generation of pups after exposure in dams is relevant because arsenic freely crosses the placenta and fetal blood barrier and can have permanent neurotoxic effects in offspring in later life [42,43,44,45]. It was reported that arsenite exposure of pregnant F0 females, only from gestational day 8 to 18, increased hepatic tumors in the F1 males of C3H mice [32,33,46]. In the present study, we exposed pregnant mice to 85 ppm NaAsO_2_ from GD 8 to 18. We have selected this period because this is one of the critical windows during that period neural tube formation, neural stem cell proliferation/differentiation and neuronal migration occurred [47].

Disruptions in social behavior and social recognition cause a variety of neuropsychiatric disorders, including depression and anxiety [48]. In this study, light-darkness transition was also tested by using light and dark chambers. This test is widely used to measure anxiety-like behavior in mice. Excessive anxiety negatively affects social life. In human, high-anxious people are more likely to avoid social interactions [24]. Studies indicated that anxiety is a good predictor of avoidance in communication with strangers [25]. Thus, we aimed to assess anxiety using light-dark transition test in the present study. Propensity to spend time in the light or dark chamber was assessed and normally they spend more time in the dark chamber. Spending more time in the light side could be interpreted as the subject mouse might be anxious [49]. In this study, time spent in the light side was not different between two groups, but mice in both groups spent more time in the light side. Observing this, we could not conclude that they were in anxious-like condition, but surely they were not in depressed mood. From the social behavioral point of view, light-dark test in the present study revealed no obvious abnormalities.

Firstly, we postulated that the impairment in social behavior might be caused due to the defect in glutamate receptors (NMDA and AMPA) which are important for synaptic plasticity, learning and memory. Social memory, in turn, is important for social behavior. Therefore, we studied various types of glutamate receptors; namely NR1, NR2A and NR2B as subunits of NMDA receptors, and GluR1, GluR2, GluR3 and GluR4 as subunits of AMPA receptors. We studied these examinations in the prefrontal cortex which is concerned with the cognitive behavior, personality expression, decision making, and moderating social behavior [29]. We found social behavior impairment but no NMDA and AMPA receptor alteration in the prefrontal cortex in 74-week-old arsenic-exposed male C3H mice. We suggested that social behavior impairment may be due to social behavior-related genes such as serotonin, inflammatory cytokines and oxidative stress in the brain other than glutamate receptors. In addition, we examined the NMDA receptor subunits (NR1, NR2A and NR2B) and AMPA receptor subunits (GluR1, GluR2, GluR3 and GluR4) in the prefrontal cortex of the control and arsenic-exposed group of 15 to 17-week-old male mice. We did not find any significant changes between the control and arsenic group. From our findings, we suggest that NMDA receptor and AMPA receptor in the prefrontal cortex may not relate to social behavior. Gestational exposure to arsenic may affect brain function permanently and social impairment was observed in 74-week-old male mice.

In contrast to the present study, Nelson-Mora et al. recently showed that there was down-regulation of AMPA receptors in hippocampus due to gestational arsenic exposure [27]. In the previous study done by Ramos-Chavez et al., down-regulation of NR2A (in the hippocampus) and NR2B (in both hippocampus and cortex) were noted [28]. The CD-1 mice were used in both studies and they were given 20 ppm of arsenic in water during the whole gestational period. Interestingly, Nelson-Mora et al. studied only in the hippocampus tissue while Ramos-Chavez et al. studied in both hippocampus and cortex. Moreover, Ramos-Chavez et al. studied in both male and female mice and the data discussed here is only male data (to be comparative to our study of C3H male mice). Ramos-Chavez et al. studied not only the prenatal exposure but also the postnatal exposure until PND15 and PND90. Although there was down-regulation of NR2A and NR2B in PND15 group, surprisingly there was up-regulation of these receptors, especially NR2A, in PND90 group compared to the control group. They explained that down-regulation might be temporary if it was not due to epigenetic changes [28]. 

Due to the negative neurochemical finding Experiment 1, we did the second experiment with the same design in 15 to 17-week-old mice to detect the possible mechanisms of the sociability impairment found in our first experiment. In our Experiment 2, we planned to study the potential neurological markers that are vulnerable to the effect of developmental arsenic exposure. It included serotonin receptor, BDNF, oxidative and inflammatory markers. In many brain regions including the prefrontal cortex, serotonin induces a decrease of glutamate transmission and a parallel increase in gamma-aminobutyric acid (GABA) transmission. These different actions of serotonin are mediated by different serotonin receptors; mostly they are inhibitory and some are excitatory. The overall action of serotonin is modulating action and serves as a “brake” on neuronal excitability. When the modulating action of serotonin is impaired, there will be hyper-excitation in pyramidal cells and excitotoxic injury will occur. These neuromodulating effects are concerned with cognitive function in the prefrontal cortex. Cognitive function is the basis of social behavior and it is called social memory [23]. The role of serotonin in prefrontal cortex is also important for normal mood. Low serotonin level is concerned with depression or anxiety while high level is responsible for anxiety disorder. If there is no serotonin, there will be severe depression and suicidal tendency [31].

Our interest in the second experiment was the serotonin, and so we studied the gene expression of the 5-HT 5B receptor in the prefrontal cortex. Among serotonin receptors, the 5-HT 5 is inhibitory subtype and some evidence suggested that blockade of 5-HT 5A receptor impairs the short-term as well as long-term memory [50]. The 5-HT 5A and 5-HT 5B are very similar to each other, but more studies are needed to explore the role of 5-HT 5B receptor. The 5-HT 5B receptor has been studied in rat hippocampus by previous studies, showing its relation to social behavior. Some researchers believed that 5-HT 5B is an autoreceptor and its up-regulation may lead to decreased extracellular serotonin and vice versa. Maekawa et al. reported that social isolation stress induced the up-regulation of 5-HT 5B receptors in the dorsal raphe nuclei of C57BL/6 mice, resulting in decreased serotonin level and abnormal behavior [51,52].

In the present study, the mRNA expression of 5-HT 5B was significantly decreased in the arsenic-exposed group. We postulated that decreased 5-HT 5B receptors would lead to decreased neuromodulating effect on glutamate transmission, resulting in hyper-excitation of pyramidal cells and cell destruction. It might be responsible for impaired cognitive function as well as impaired social behavior. Regarding autoreceptor action of 5-HT 5B, extracellular serotonin level might be increased in that case. The arsenic-exposed mice in the present study showed impaired social behavior, but they were not in depressed mood. This situation was in accordance with the decreased 5-HT 5B expression. In contrast, Martinez et al. reported that C57BL/6J mice showed depressive-like behavior with increased sensitivity of 5-HT 1A in the hippocampus as an effect of perinatal arsenic exposure (50 ppb) [20].

For synaptogenesis and synaptic plasticity, BDNF is also an important factor. Moreover, BNDF interacts with serotonin; BDNF promotes the development and function of serotonergic neurons, and serotonergic transmission exerts powerful control over BDNF expression [30]. In this study, gene expression for BDNF was decreased in the arsenic-exposed group. Decreased BDNF expression during brain development might impair the development of neurons in the prefrontal cortex, resulting in dysfunction of the prefrontal cortex. Impaired social behavior might be due to impaired synaptogenesis and synaptic plasticity necessary for learning and memory (direct effect of BDNF) as well as impaired serotonergic development and function (indirect effect of BDNF).

In most of the pathophysiological processes of arsenic toxicity, oxidative stress and inflammatory process are usually involved [13]. Therefore, we studied the gene expression of HO-1 as an oxidative stress marker and that of COX-2 as the potent inflammatory marker. However, there were no significant changes of these two markers in the present study. These findings were in accordance with that of the study done by Ramos-Chavez et al. in which glutathione (GSH) synthesis was increased (due to oxidative stress) only in PND1 group and it returned to normal in PND15 group [28]. Arsenic toxicity might induce oxidative stress at the time of exposure (gestational period), but there was no evidence of oxidative stress or inflammation in adult life (at the time of the present study).

## 5. Conclusions

In the prefrontal cortex, serotonin plays an important role by modulating the excitatory glutamate transmission from pyramidal cells and inhibitory GABA transmission from interneurons. This modulating action is critical for cognitive function, and its impairment leads to memory defect and abnormal behavior. Different types of serotonin receptors contribute to this neuromodulating action, balancing each other, but the detailed mechanism is still disputed up to now. BDNF also interact with serotonergic system, contributing to cognitive function. Cognitive function is the basis for social memory and social behavior.

In the present study, F1 adult male mice showed poor sociability and social novelty preference. Decreased gene expressions of 5-HT 5B and BDNF were also detected. Changes in oxidative stress marker and inflammatory marker were not detected in the present study. Taken together, these findings suggest that developmental arsenic exposure might affect social memory and social behavior by disturbing the serotonergic system and reducing BDNF in the prefrontal cortex.

## Figures and Tables

**Figure 1 ijerph-16-00174-f001:**
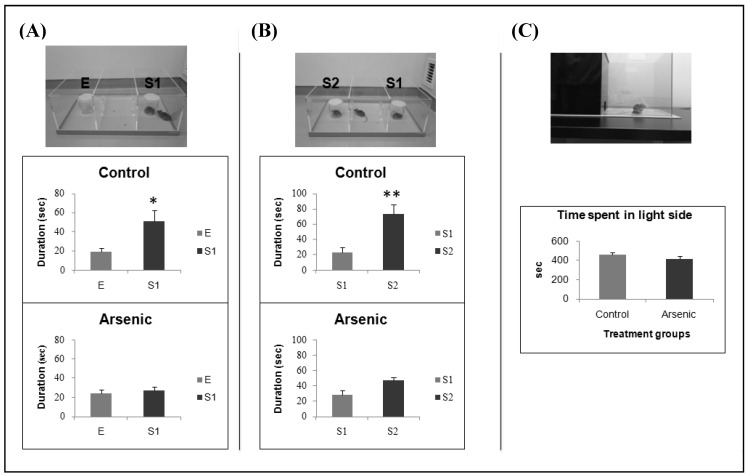
Effect of developmental arsenic exposure on social behavior: (**A**) Sociability test, (**B**) Social novelty preference test, and (**C**) Light-dark transition test in the control group and the arsenic-exposed group of 74-week-old male mice. Each bar represents the mean ± SE *(n* = 14, ** p* < 0.05, *** p* < 0.001). E, empty; S1, stranger 1; S2, stranger 2.

**Figure 2 ijerph-16-00174-f002:**
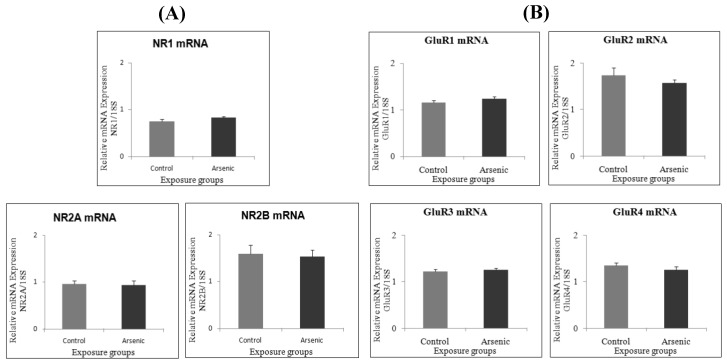
Messenger RNA expression level of (**A**) NMDA receptor subunits, and (**B**) AMPA receptor subunits in the prefrontal cortex of the control group and the arsenic-exposed group of 74-week-old male mice. Each bar represents the mean ± SE (*n =* 14).

**Figure 3 ijerph-16-00174-f003:**
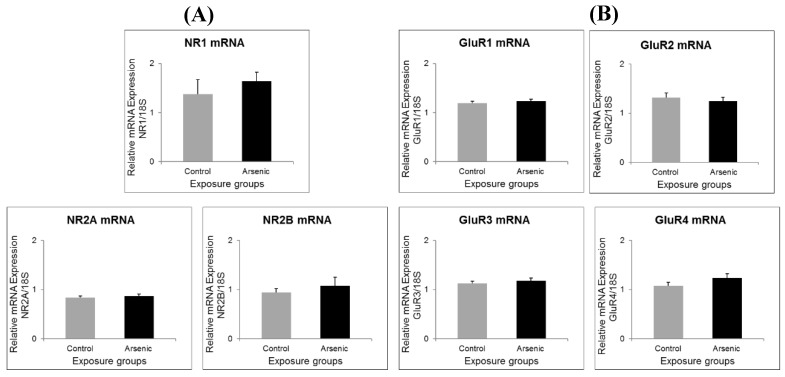
Messenger RNA expression level of (**A**) NMDA receptor subunits, and (**B**) AMPA receptor subunits in the prefrontal cortex of the control group and the arsenic-exposed group of 15–17-week-old male mice. Each bar represents the mean ± SE (*n* = 7).

**Figure 4 ijerph-16-00174-f004:**
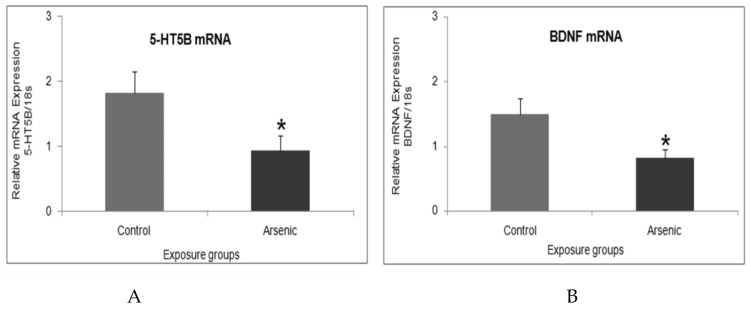
Messenger RNA expression level of (**A**) 5-HT 5B, (**B**) BDNF, (**C**) HO-1 and (**D**) COX-2 in the prefrontal cortex of the control group and the arsenic-exposed group of 15–17-week-old male mice. Each bar represents the mean ± SE (*n* = 7, ** p* < 0.05).

**Table 1 ijerph-16-00174-t001:** Comparison of animal studies on the neurotoxic effects of developmental arsenic exposure.

Study	Animal	Level of Exposure	Time of Exposure	Observed Findings
Rodriguez et al., 2002 [19]	Sprague–Dawley rats	36.7 ppm	GD15-PND120 PND1-PND120	Increased number of errors in a delayed alternation task.
Aung et al., 2016 [22]	C3H mice	85 ppm	GD8-GD18	Behavioral inflexibility.
Present study, 2018	C3H mice	85 ppm	GD8-GD18	Impaired sociability and social novelty preference.

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
