# Peer review of "Effects of Developmental Arsenic Exposure on the Social Behavior and Related Gene Expression in C3H Adult Male Mice"

_ijerph, 2019, doi:10.3390/ijerph16020174_

Round 1

Reviewer 1 Report

In this article, adult pregnant female mice were exposed to 85ppm arsenic during gestational day 8 to 18. F1 male pups (74 and 15-17 weeks old) were tested for social behavior and social behavior-related genes, as well as oxidative stress and inflammation-related genes (only in younger mice). 

In the introduction section, the authors mentioned that arsenic is a naturally occurring metalloid, but they do not indicate whether arsenic exposure is a problem and which populations are at risk of intoxication, this is important because mice are exposed to a high concentration of arsenic during development and it is not clear if this is a “realistic” concentration under the context of public health. In addition, the authors should explain why only male pups were used for both experiments.

The authors performed two experiments, both of them followed the same experimental design and protocols, the only difference was that the first experiment used older mice (74 weeks old) while the second experiment used younger mice (15-17 weeks old), in both cases social behavior was evaluated but different genes were measured. It is not clear why the authors did not measure the expression of the same genes in both experiments; the same RNA would have been enough to measure expression of all genes, and both age groups could have been compared. The need for younger mice for experiment 2 should be explained in the Materials and Methods section. 

For qPCR analysis, the authors used only one reference gene (18S rRNA), it is necessary to explain if a stability analysis was performed with other reference genes, the 18S rRNA gene is not always the best option and may affect the interpretation of data.

Since protein data are not available in this study, be cautious when discussing results based only on gene expression data, since transcript levels may not reflect the amount of functional protein.

Some English edition is necessary (e.g. lines 142-148), use “gene expression” instead of “gene expressions”. 

Author Response

Responses to the Reviewer 1

In this article, adult pregnant female mice were exposed to 85ppm arsenic during gestational day 8 to 18. F1 male pups (74 and 15-17 weeks old) were tested for social behavior and social behavior-related genes, as well as oxidative stress and inflammation-related genes (only in younger mice).

In the introduction section, the authors mentioned that arsenic is a naturally occurring metalloid, but they do not indicate whether arsenic exposure is a problem and which populations are at risk of intoxication, this is important because mice are exposed to a high concentration of arsenic during development and it is not clear if this is a “realistic” concentration under the context of public health. In addition, the authors should explain why only male pups were used for both experiments.

  As commented by the Reviewer, we added arsenic problem in human as follows; Arsenic is a naturally occurring metalloid and can be found in water as the inorganic form (arsenite or arsenate) (Hughes et al., 2011). Arsenic contamination in drinking water is a worldwide problem, especially in Bangladesh. According to the study of Ahmad et al., 50 million people in Bangladesh are exposed to arsenic-contaminated water (above 0.05mg/L) and the highest level of contamination is 2.97mg/L (Ahmad et al., 2001). There are several health problems related to arsenic exposure; skin lesions, cardiovascular disease, neurological disorders, and diabetes (ATSDR, 2007; Htway et al., 2014; Naing, 2017). We provide that issue in Introduction section in our revised manuscript.

 For selection of 85 ppm NaAsO2 dose was mentioned as follows in Materials and Methods section.

The reasons for selection of dose of NaAsO2 85 ppm are 1) species difference was observed between human and mouse and high dose is required for detection of neurotoxic effects in C3H mice and no maternal toxicity and teratogenicity were not observed at the dose of present study; 2) this is a standard dose to detect the effects of developmental exposure to arsenic on neurotoxicity, reproductive toxicity and carcinogenicity in our research group (Aung et al., 2016, Nohara et al., 2012; 2017).

 The reasons for using only male pups are 1) there is a sex difference in susceptibility to arsenic; 2) there is estrous cycle in female and possible hormonal effect on social behavior (Rodriguez et al., 2002).

The authors performed two experiments, both of them followed the same experimental design and protocols, the only difference was that the first experiment used older mice (74 weeks old) while the second experiment used younger mice (15-17 weeks old), in both cases social behavior was evaluated but different genes were measured. It is not clear why the authors did not measure the expression of the same genes in both experiments; the same RNA would have been enough to measure expression of all genes, and both age groups could have been compared. The need for younger mice for experiment 2 should be explained in the Materials and Methods section.

   In fact, the previous study showed that behavioral inflexibility and cortical disarrangement in prelimbic cortex was observed in 60-week-old C3H male mice after prenatal exposure to 85 ppm arsenic (Aung et al., 2016). This finding prompted us to examine social behavior after same exposure period (GD 8 to 18). Therefore, we performed same exposure design and examined social behavior and glutamate receptor NMDA and AMPA in the prefrontal cortex in 74-week-old male C3H mice (Experiment 1 in the present study). We found social behavior impairment but no NMDA and AMPA receptor alteration in the prefrontal cortex in 74-week-old male C3H mice. We suggested that social behavior impairment may be due to social behavior-related genes such as serotonin, inflammatory cytokines and oxidative stress in the brain other than glutamate receptors. Therefore, we aimed to study the effect of gestational exposure to 85ppm arsenic on social behavior and their related gene expressions in the prefrontal cortex in 15-17-week-old C3H male mice. Unfortunately, we could not perform social behavior test and only examined social behavior related gene expressions only in the prefrontal cortex. We provide that issue in Materials and Methods section in our revised manuscript.

For qPCR analysis, the authors used only one reference gene (18S rRNA), it is necessary to explain if a stability analysis was performed with other reference genes, the 18S rRNA gene is not always the best option and may affect the interpretation of data.

    We selected 18S rRNA is a reference gene because forward and reverse primer designs were made ourselves and we have tested many times for its stability.  

Since protein data are not available in this study, be cautious when discussing results based only on gene expression data, since transcript levels may not reflect the amount of functional protein.

     We agree the Reviewer’s comment and we have a plan to do Western blot analysis of proteins related to social behavior in our future study.

Some English edition is necessary (eg., lines 142-148), use “gene expression” instead of “gene expressions”.

     As commented by the Reviewer, we corrected “gene expressions” to “gene expression” in our revised manuscript. Then, we also made English editing of our revised manuscript thoroughly throughout the manuscript.

Reviewer 2 Report

The statement “C3H mice are good for neurological studies” in line 179 should be supported with some concrete reference studies, signifying their importance in the present study.

The rationale behind dose selection for arsenic exposure should be clearly mentioned to bring forth the severity of its effects more accurately.

A description of human-relevant conditions relatable to the experimental conditions used in the study would shed more light over its overall translational value.

The normal drinking water also contains smaller or larger amounts of arsenic depending upon the source and location from where water has been taken. An estimation of the same would prevent a possible bias in the study.

Can the authors elaborate if all the dams under study were from the same mother and how many mice were kept per cage while weaning and while carrying out a social interaction test?

As per the text, there seems to be an ambiguity in the experimental paradigms that were followed. For example, were the mice used for behavioral studies sacrificed for neurochemical estimation or was some other set of mice used for the latter?

Studies have indicated that arsenic crosses the BBB as well as placental barrier. Therefore, keeping in view that the metabolism of arsenic varies a lot, it is always advisable to assess the amount of arsenic that crosses the placental barrier and reaches the brain of the pups. This will help to draw more conclusive results.

The authors have included light dark chamber test in the social interaction tests. Can they please elaborate how this will serve as a social interaction test and what is its exact purpose here?

The details of primer sequence are missing.

Line numbers 197-200 represent a generalized statement that doesn’t really go with the discussion portion and hence requires a little bit more clarification.

The experimental design for social behavior assessment needs some more clarity. Were the social behavior and social novelty preference tests conducted during the same or at different times?

The time points used in the study for experiment 1 and 2 are different, i.e., 74 weeks and 16 to 17 weeks respectively. What was the exact rationale behind this?

The negative findings of the study need a clearer and more proper justification for indicating their significance in the present study.

As per the graphical representation levels of COX-2 seem to be varying enough to have a significant p-value. However, even if it is non-significant there is a substantial decrease in the expression of COX-2 which can be further explained in the context of the present study.

The endpoints chosen for experiment 1 and 2 need to be supported more strongly with logical reasoning to bring out their significance in the present study.

The discussion portion demands more focus and emphasis on the findings of the current study, supported with relevant references so that the authors can highlight their own key findings instead of earlier studies.

The chosen duration for arsenic exposure in mothers and its subsequent effect over F1 generation of pups lacks a concrete conceptual backing.?

Could the authors please elaborate regarding the overall result of the study? According to the present study exposure of arsenic from (GD) 8 to (GD) 18 in pregnant resulted in decreased expression of 5HT5B, BDNF, HO1, and COX-2 mRNA in 15-17 week old mice offspring, even though the offspring were not exposed to arsenic. No difference in the mRNA levels of NR1, NR2A, NR2B, GluR1, GluR2, GluR3 and GluR4 was observed at 74 weeks. Is it possible that this trend could be due to the elimination of arsenic from the offspring? Please justify.

Furthermore, it would be more meaningful if authors could conduct western blotting of the proteins associated with social behavior to give a more concrete result.    

The authors can get redundancy and grammatical errors in the entire manuscript checked again.

Author Response

Responses to the Reviewer 2

The statement “C3H mice are good for neurological studies” in line 179 should be supported with some concrete reference studies, signifying their importance in the present study.

    As suggested by the Reviewer, we add relevant reference and mention for selection in the present study as follows in our revised manuscript.

In the present study, C3H mice were used because C3H mice are good for behavior assessment and neurochemical analysis (Cases et al.,1995).

The rationale behind dose selection for arsenic exposure should be clearly mentioned to bring forth the severity of its effects more accurately

   The reason for selection of dose of NaAsO2 85 ppm are 1) species difference was observed between human and mouse and high dose is required for detection of neurotoxic effects in C3H mice and no maternal toxicity and teratogenicity were not observed at the dose of present study; 2) this is a standard dose to detect the effects of developmental exposure to arsenic on neurotoxicity, reproductive toxicity and carcinogenicity in our research group (Aung et al., 2016, Nohara et al., 2012; 2017). We provide that issue in Materials and Methods section in our revised manuscript.

A description of human-relevant conditions relatable to the experimental conditions used in the study would shed more light over its overall translational value.

   As suggested by the Reviewer, we add human relevant condition as follows in Introduction section in our revised manuscript as follows;

Epidemiological studies have indicated that exposure to arsenic in drinking water in utero or early childhood resulted in increase of mortality in young adults by bladder cancer, laryngeal cancer, lung cancer and chronic renal disease (Smith et al., 2006; 2012; Steinmaus et al., 2016).

The normal drinking water also contains smaller or larger amounts of arsenic depending upon the source and location from where water has been taken. An estimation of the same would prevent a possible bias in the study.

 As commented by the Reviewer, normal drinking water also contains smaller or larger amounts of arsenic depending upon the source and location from where water has been taken. In the present study, we used tap water from our research Institute as a control and added arsenic in tap water for exposure group. We provide that issue in our revised manuscript.

Can the authors elaborate if all the dams under study were from the same mother and how many mice were kept per cage while weaning and while carrying out a social interaction test?

  We have already mentioned in our manuscript as F1 generation of male pups (n=14) from different mothers. We described that issue in our materials and Methods section in our revised manuscript as follows;

  The F1 male pups were weaned at postnatal day 21 and were randomly selected from different dams and housed under the same condition as the dams (n=3 per cage). At the age of 70-week, male pups were housed in individual cage. Social behavior task was examined at the age of 74-week-old using three-chamber social behavior apparatus.

As per the text, there seems to be an ambiguity in the experimental paradigms that were followed. For example, were the mice used for behavioral studies sacrificed for neurochemical estimation or was some other set of mice used for the latter?

  In the present study, for experiment (1), after completion of behavior test, 74-week-old male mice were sacrificed under deep anesthesia and collected prefrontal cortex for gene expression assay. For experiment (2), without doing behavior test, the 15 to17-week-old male mice were sacrificed under deep anesthesia and collected prefrontal cortex for gene expression assay. We provide this issue in materials and Methods section in our revised manuscript.

Studies have indicated that arsenic crosses the BBB as well as placental barrier. Therefore, keeping in view that the metabolism of arsenic varies a lot, it is always advisable to assess the amount of arsenic that crosses the placental barrier and reaches the brain of the pups. This will help to draw more conclusive results.

 As suggested by the Reviewer, total arsenic (TA) should be measured in a study in which arsenic exposure is given continuously till the time of tissue study (like the study of Rodriguez et al., 2002). In the present study, exposure was given during gestation and tissue study was done only at the adulthood. According to the study done by Markowski et al. (2010), TA in brain was detected only in PND1 mice and cleared in PND21 mice after gestation exposure to arsenic (up to 100ppm). (Markowski VP, Currie D, Reeve EA, Thompson D, Wise Sr JP. Tissue-specific and dose-related accumulation of arsenic in mouse offspring following maternal consumption of arsenic-contaminated water. Basic & Clinical Pharmacology & Toxicology. 2010;108(5):326-32.)

The authors have included light dark chamber test in the social interaction tests. Can they please elaborate how this will serve as a social interaction test and what is its exact purpose here?

Light dark transition test is used for anxiety assessment. Excessive anxiety negatively affects social life. In human, high-anxious people are more likely to avoid social interactions (Wu et al., 2013). Studies indicated that anxiety is a good predictor of avoidance in communication with strangers (Duronto et al., 2005). Thus, we aimed to assess anxiety using light dark transition test in the present study. We add this issue in Discusion section in our revised manuscript.

The details of primer sequence are missing.

The primers (NR1, NR2A, NR2B, GluR1, GluR2, GluR4, serotonin receptor 5HT5B, BDNF, HO1, COX2 were purchased from Qiagen, Sample and Assay Technologies (they do not provide primer sequences) and we provided NM number. We designed 18S forward and reverse primers and purchased from Hokkaido System Science (Hokkaido, Japan) and we expressed primer sequence in Materials and Methods section.

Line numbers 197-200 represent a generalized statement that doesn’t really go with the discussion portion and hence requires a little bit more clarification.

  As commented by the Reviewer, we re-write that sentence as follows;

Aung et al., also reported that gestation exposure (GD 8 to 18) to 85 ppm NaAsO2 resulted in increased discrimination error rate in spatial learning activity tested by IntelliCage accompanied with neuronal morphology changes in 60-week-old male mice.

The experimental design for social behavior assessment needs some more clarity. Were the social behavior and social novelty preference tests conducted during the same or at different times?

   As commented by the Reviewer, we mentioned social behavior test as follows in Materials and Methods section in our revised manuscript.

   Social behavior test consists of 3 phases including habituation phase for 5 min, sociability phase for 10 min and social novelty preference for 10 min.

The time points used in the study for experiment 1 and 2 are different, i.e., 74 weeks and 16 to 17 weeks respectively. What was the exact rationale behind this?

   In fact, our previous study showed that behavioral inflexibility and cortical disarrangement in prelimbic cortex was observed in 60-week-old C3H male mice after prenatal exposure to 85 ppm arsenic (Aung et al., 2016). This finding prompted us to examine social behavior after same exposure period (GD 8 to 18). Therefore, we performed same exposure design and examined social behavior and glutamate receptor NMDA and AMPA in the prefrontal cortex in 74-week-old male C3H mice (Experiment 1 in the present study). We found social behavior impairment but no NMDA and AMPA receptor alteration in the prefrontal cortex in 74-week-old male C3H mice. We suggested that social behavior impairment may be due to social behavior-related genes such as serotonin, inflammatory cytokines and oxidative stress in the brain other than glutamate receptors. Therefore, we aimed to study the effect of gestational exposure to 85 ppm arsenic on social behavior and their related gene expressions in the prefrontal cortex in 15-17-week-old C3H male mice. Unfortunately, we could not perform social behavior test and only examined social behavior related gene expressions only in the prefrontal cortex. We provide this issue in materials and Methods section in our revised manuscript.

The negative findings of the study need a clearer and more proper justification for indicating their significance in the present study.

   As suggested by the Reviewer, we mentioned our negative findings as follows in Discussion section in our revised manuscript.

We found social behavior impairment but no NMDA and AMPA receptor alteration in the prefrontal cortex in 74-week-old arsenic-exposed male C3H mice. We suggested that social behavior impairment may be due to social behavior-related genes such as serotonin, inflammatory cytokines and oxidative stress in the brain other than glutamate receptors.

As per the graphical representation levels of COX-2 seem to be varying enough to have a significant p-value. However, even if it is non-significant there is a substantial decrease in the expression of COX-2 which can be further explained in the context of the present study.

   Regarding COX-2 mRNA, it seems decreased significantly, however, not statistically significant. Thus, we mentioned as “potent inflammatory marker COX-2 mRNA tended to decrease in arsenic-exposed group, but not statistically significant.” in our revised manuscript. 

The endpoints chosen for experiment 1 and 2 need to be supported more strongly with logical reasoning to bring out their significance in the present study.

 The endpoints chosen for experiment 1 and 2 were mentioned in above comment.

The discussion portion demands more focus and emphasis on the findings of the current study, supported with relevant references so that the authors can highlight their own key findings instead of earlier studies.

    As commented by the Reviewer, we added discussion based on our key findings in Discussion section in our revised manuscript.

The chosen duration of arsenic exposure in mothers and its subsequent effect over F1 generation of pups lacks a concrete conceptual backing?

   It was reported that early life arsenic exposure in rodents can affect higher brain functions permanently through later life. Critical periods may be gestational, perinatal and early childhood (Kozul-Horvath et al., 2012; Aung et al., 2016). Investigation of F1 generation of pups after exposure in dams is relevant because arsenic freely crosses the placenta and fetal blood barrier and can have permanent neurotoxic effects in offspring in later life (Chaineau et al., 1990; Li et al., 2009; Rios et al, 2009; Chandravanshi et al., 2014). It was reported that arsenite exposure of pregnant F0 females, only from gestational day 8 to 18, increased hepatic tumors in the F1 males of C3H mice (Waalkes et al., 2003; Nohara et al., 2012; 2017). In the present study, we exposed pregnant mice to 85 ppm NaAsO2 from GD 8 to 18. We have selected this period because this is one of the critical windows during that period neural tube formation, neural stem cell proliferation/differentiation and neuronal migration occurred (Alfonso-Loeches and Consuelo Guerri, 2011). We add this issue in Discussion section in our revised manuscript.

Could the authors please elaborate regarding the overall result of the study? According to the present study exposure of arsenic from (GD) 8 to (GD) 18 in pregnant resulted in decreased expression of 5HT5B, BDNF, HO1, and COX-2 mRNA in 15-17 week old mice offspring, even though the offspring were not exposed to arsenic. No difference in the mRNA levels of NR1, NR2A, NR2B, GluR1, GluR2, GluR3 and GluR4 was observed at 74 weeks. Is it possible that this trend could be due to the elimination of arsenic from the offspring? Please justify

  We have examined the NMDA receptor subunit NR1, NR2A and NR2B and AMPA receptor subunit GluR1, GluR2, GluR3 and GluR4 in the prefrontal cortex of the control and arsenic-exposed group of 15 to 17-week-old male mice. We do not find any significant changes between the control and arsenic group. From our findings we suggest that NMDA receptor and AMPA receptor in the prefrontal cortex may not relate to social behavior. Gestational exposure to arsenic may affect brain function permanently and social impairment was observed in 74-week-old male mice.   

   We added this issue in Results and Discussion section in our revised manuscript.

Furthermore, it would be more meaningful if authors could conduct western blotting of the proteins associated with social behavior to give a more concrete result.

   We agree the Reviewer’s comment and we have a plan to do Western blot analysis of proteins related to social behavior in our future study.

The authors can get redundancy and grammatical errors in the entire manuscript checked again.

   As suggested by the Reviewer, we made English editing of our revised manuscript thoroughly throughout the manuscript.

Round 2

Reviewer 1 Report

In this revised version the authors answered all my comments and suggestions, I believe this is an improved version of the manuscript. 

In the Conclusion section, lines 373-374 "these findings suggest" instead of "suggested". 

Author Response

Responses to the Reviewer 1

In this revised version the authors answered all my comments and suggestions. I believe this is an improved version of the manuscript.

In the Conclusion section, lines 373-374 “these findings suggest” instead of “suggested”.

As commented by the Reviewer, we corrected “suggested” to “suggest” in our revised manuscript. We also made English editing in our revised manuscript. A thorough grammatical proofread was done by the Head of the English Department, University of Medicine, Magway.

Reviewer 2 Report

THE ENTIRE MANUSCRIPT STILL NEEDS A THOROUGH GRAMMATICAL PROOF READ

Line 31: Keeping in mind the ever changing scenario and numbers, Ahmad et al., 2001 is an old reference to cite to report the present situation.

Line 63-68: It would be better if the authors could justify the rationale and novelty addressed for conducting this study more clearly.

Estimating the amount of arsenic in the “tap water” administered to the mice would be a better option since it too might have some amount arsenic in it and this could change the amount administered to the mice i.e. 85 ppm. The same goes to the amount of arsenic that might be administered to the control “unknowingly” through tap water.

Which anesthetic was utilized to produce deep anesthesia? Does this anesthetic cause any interaction with AMPA receptors or NMDA receptor under study?

Authors can categorize the material and method section more efficiently. Information regarding preparation of dose - sodium arsenite etc. should be mentioned in a separate heading. In addition the information pertaining to mice and other ethical permissions should be mentioned in one column. This will help the potential readers in understanding the methodology more clearly.

An insight regarding the use of prefrontal cortex in the present study would be very beneficial.

As indicated by the authors “Light dark transition test is used for anxiety assessment” is completely justified, however, in the material and method section it should be mentioned under different heading and not in social interaction test.

Could authors please elaborate, how change in mRNA expression (and not protein) of serotonin gene at 15-16 week affect behavior at 74 weeks? The authors have already clarified that they did not conduct behavior assessment in experiment 2, however, it would be better if the authors could conduct neurochemical experiment including qPCR for genes such as serotonin, and inflammatory cytokines and oxidative stress to address the neurobehavior issue more clearly at 74 weeks.

Line 225-232 The emphasis is more on the previous studies than the author’s own hypothesis and result.

Overall, the discussion section has repetition of already explained results at many places. It is expected that discussion should accompany reasoning as to what and why the results are.

Line 306-309: Noncoherent text.

References missing in many places. Please check.

Author Response

Responses to the Reviewer 2

THE ENTIRE MANUSCRIPT STILL NEEDS A THOROUGH GRAMMATICAL PROOF READ.

As suggested by the Reviewer, a thorough grammatical proofread was done by the Head of the English Department, University of Medicine, Magway.

Line 31: Keeping in mind the ever changing scenario and numbers, Ahmad et al., 2001 is an old reference to cite to report the present situation.

As suggested by the Reviewer, we added the update data of Bangladesh as follows in our revised manuscript.

In 2012, it was estimated that about 19 million people in Bangladesh were still exposed to arsenic concentration above the national standard of 0.05mg/L. According to the World Health Organization provisional guideline value (0.01mg/L), about 39 million people in Bangladesh were exposed to arsenic-contaminated water (BBS/UNICEF, 2015).

Line 63-68: It would be better if the authors could justify the rationale and novelty addressed for conducting this study more clearly.

As suggested by the Reviewer, we re-write the justification as follows in our revised manuscript.

It still needs to explore the mechanisms of developmental neurotoxicity and behavioral alterations induced by arsenic exposure. Little is known about the effect of arsenic exposure during brain development on social behavior. Social behavior is related to learning and memory (Ciranna, 2006). It is also influenced by mood changes (Kaidanovich-Beilin et al., 2011; Wu et al., 2013). Most of the previous studies investigated the changes in the hippocampus (Tyler & Allan, 2013; Luo et al., 2009; Nelson-Mora et al., 2018; Ramos-Chavez et al., 2015). The prefrontal cortex is responsible for moderating social behavior (Fuster, 2015). This study aimed to detect the effect of developmental arsenic exposure on social behavior and related gene expressions in the prefrontal cortex of C3H adult male mice.

Estimating the amount of arsenic in the “tap water” administered to the mice would be a better option since it too might have some amount arsenic in it and this could change the amount administered to the mice i.e. 85ppm. The same goes to the amount of arsenic that might be administered to the control “unknowingly” through tap water.

Tap water in Japan is safe to drink. Arsenic is not detectable in tap water everywhere in Japan. According to Japan Water Works Association (Nihon SuidouKyokai), Japan is one of only 15 (out of 196) countries in the world with potable tap water (Global Water Supply and Sanitation Assessment 2000 Report by the World Health Organization (WHO), UNICEF, and Water Supply & Sanitation Collaborative Council).

Which anesthetic was utilized to produce deep anesthesia? Does this anesthetic cause any interaction with AMPA receptors or NMDA receptor under study?

Pentobarbital anesthesia was utilized in the present study. It had been used in our previous neurological studies (Win-Shwe, 2015; 2018) in which NMDA receptors, serotonin receptor, BDNF, and HO-1 were studied. We showed the up-regulation of NR1 (NMDA receptor) mRNA expression in one of the study groups (Win-Shwe, 2015). Negative neurochemical finding in the present study might not be due to anesthesia.

Authors can categorize the material and method section more efficiently. Information regarding preparation of dose – sodium arsenite etc. should be mentioned in a separate heading. In addition, the information pertaining to mice and other ethical permissions should be mentioned in one column. This will help the potential readers in understanding the methodology more clearly.

As suggested by the Reviewer, we added two headings in the materials and methods section as follows in our revised manuscript.

2.1.1. Animal

2.1.2. Social behavior

2.1.3. Light-dark transition test

2.1.4. Quantification of social behavior-related gene expression

An insight regarding the use of prefrontal cortex in the present study would be very beneficial.

The reason for using the prefrontal cortex was mentioned in the introduction and discussion sections as follows in our revised manuscript.

The prefrontal cortex is responsible for moderating social behavior(Fuster, 2015). (Introduction)

We studied these examinations in the prefrontal cortex which is concerned with the cognitive behavior, personality expression, decision making, and moderating social behavior (Fuster, 2015). (Discussion)

As indicated by the authors “Light dark transition test is used for anxiety assessment” is completely justified, however, in the material and method section it should be mentioned under different heading and not in social interaction test.

As suggested by the Reviewer, we mentioned “light-dark transition test” under different heading in our revised manuscript.

Could authors please elaborate, how change in mRNA expression (and not protein) of serotonin gene at 15-16 week affect behavior at 74 weeks? The authors have already clarified that they did not conduct behavior assessment in experiment 2, however, it would be better if the authors could conduct neurochemical experiment including qPCR for genes such as serotonin, and inflammatory cytokines and oxidative stress to address the neurobehavior issue more clearly at 74 weeks.

According to the literature, gestational exposure to arsenic may affect brain function permanently and behavioral alteration would be observed through later life (Kozul-Horvath et al., 2012). In the study of Rodriguez et al., behavioral alteration was found in 16-17-week-old mice. Similarly, behavioral inflexibility was detected in 60-week-old male mice of the study of Aung et al. (Rodriguez et al., 2002; Aung et al., 2016). In our study, we found impaired social behavior in 74-week-old mice (experiment 1). Unfortunately, behavior task could not be performed in 15-17-week-old mice (experiment 2). We supposed that impaired social behavior would be also found in the 15-17-week-old mice if we had a chance to do the behavior task in the experiment 2. We assumed that it does not matter whatever the age is. Therefore, we believed that qPCR results of the 15-17-week-old mice in the experiment 2 could be linked up with the experiment 1 data (impaired social behavior in 74-week-old mice).

We already performed the qPCR for glutamate receptors in 15-17-week-old mice and we proved that the results of the experiment 2 were very similar to that of the experiment 1. Actually, we did the experiment 2 as a supplementary to the experiment 1, exploring new neurochemical factors. Unfortunately, there was a difference in age of the mice between the experiment 1 and 2. We did not intend to compare two age groups in our study. Repeating the same neurochemical study was not our primary objective. We will consider protein study or epigenetic study in our future plan.

Line 225-232: The emphasis is more on the previous studies than the author’s own hypothesis and result.

As suggested by the Reviewer, we dropped the unnecessary lines in our revised manuscript.

In contrast, their study was done in Sprague–Dawley rats by giving low dose arsenic. They divided into 3 groups; one group (n=4) was the exposed group (starting from GD15), the other group (n=4) was also the exposed group (starting from PND1), and the control group (n=4) given distilled water as drinking water.

Overall, the discussion section has repetition of already explained results at many places. It is expected that discussion should accompany reasoning as to what and why the results are.

As suggested by the Reviewer, we revised the whole discussion section.

Line 306-309: Noncoherent text

As suggested by the Reviewer, we dropped the non-coherent text in our revised manuscript.

In our study, we gave arsenic in drinking water during gestation (GD8-GD18) and studied only when the mice were 74-week-old, and so the negative findings of our study might be due to the time of study.

References missing in many places. Please check.

As suggested by the Reviewer, we added references in necessary places in our revised manuscript.
